# Evaluation of Technical Efficiency in Exotic Carp Polyculture in Northern India: Conventional DEA vs. Bootstrapping Methods

Hongzhi Zhang [1], Ubair Nisar [2,*] and Yongtong Mu [2,*]

1 Research and Development Department, Shandong Foreign Trade Vocational College, Qingdao 266100, China; ouqdsnow@163.com
2 Key Laboratory of Mariculture (Ministry of Education), Fisheries College, Ocean University of China, No. 5 Yushan Road, Qingdao 266003, China
* Correspondence: ubairnisar@stu.ouc.edu.cn (U.N.); ytmu@ouc.edu.cn (Y.M.); Tel.: +86-17866854967 (U.N.); +86-13156296999 (Y.M.)

**Abstract:** The paper adopts the conventional data envelopment analysis (DEA) and bootstrap procedure to analyze the technical efficiency, while tobit regression is applied to identify the factors affecting efficiencies of exotic fish polyculture in Jammu and Kashmir (India). According to the statistical analysis of the variables utilized, there was a lot of variability in the inputs being used by the farmers, with the most variation being in the lime input. The DEA estimated technical efficiency for the sample farms in Jammu and Kashmir is 0.9771 and 0.9741, respectively, with least technical inefficiency of 3%. The bias-corrected (bootstrapped) technical efficiencies found were slightly lower than the ones estimated by conventional DEA. Mean allocative and cost efficiencies for sample farms in the Jammu region were 92% and 75%, respectively, and 84% and 74%, respectively, for farmers in the Kashmir region. Farming experience, age, and education have favorable impacts on technical efficiency of farmers in the state, but family size showed negative impacts. Efficiency improvement will eventually lead to increase in the production providing better scope for marketed surplus. More fisheries extension is suggested for expanding the exotic fish culture in the state making the enterprise a more profitable venture.

**Keywords:** aquaculture; economic efficiency; data envelopment analysis; bootstrapping tobit regression; inefficiency; India

## 1. Introduction

Feeding a burgeoning population, which is anticipated to reach 9.6 billion by 2050, is a major concern for the world. Food production must expand in a world where resources for food production, such as land and water, are becoming scarcer in a more populated globe; the world must adapt how it undertakes economic activities in the light of anthropogenic climate change [1]. As a result, one of the key issues in providing food for the world's rising population is efficient input allocation in food production. In India, capture and culture fisheries provide a livelihood and profitable employment to more than 14 million people and they also provide nutritional security. Fisheries contribute about 1.07% to the GDP of the country and 5.23% to the Agricultural GDP [2]. In Asia, aquaculture is dominated by the major carp and usually contributes around 80% of the total aquaculture production [3]. The cultured major carp, to a greater extent, consist of Indian major carp and exotic carp. Catla (*Catla catla*), Rohu (*Labeo rohita*), and Mrigal (*Cirrhinus mrigala*) are the three principal carp species used in freshwater aquaculture in India. Because of their explosive growth and consumer acceptance, major carp are the most popular farmed fish [4].

The study was carried out in Jammu and Kashmir located in northern India. In the region, the agriculturists initially embraced the exotic carp (silver carp, common carp, and grass carp) polyculture as a supplementary endeavor to horticulture for expanding their earnings. It is observed that exotic carp culture in the state contributes to around 70% of

the total aquaculture but the efficiency is far below the national average [5]. Fish are an important food item and those who dwell near lakes and the floating population of boatmen depend on fishery activities for a considerable part of their diet. Effective farms either create more yield than others for a given set of inputs or produce a given yield with the lowest level of input. Therefore, in order to maximize the economic returns, the primary goal should be to utilize the existing resources in the best possible manner. In Jammu and Kashmir, the farmers practice polyculture of exotic carp such as silver carp (*Hypophthalmichthys molitrix*), common carp (*Cyprinus carpio*), and grass carp (*Ctenopharyngodon idella*). The mixed stocks of these fish species do not compete with food and have different ecological requirements. This enhances the fish production for given input quantitates and increases the efficiency of the farmers. The exotic carp farming in the region is a profitable venture and must be employed and promoted for gainful employment of the younger generation [6].

There is a huge demand for fish in the market, and farmers must either embrace new technology or improve their utilization of input resources to bridge the gap between demand and supply. This can be supplemented by analyzing the factors affecting the technical efficiency of the sample fish farms of the state and further improving them. Understanding the efficiency status is exceptionally significant for culture management and policy implications. Research can further reveal the probability of increasing the productivity by identifying the constraints and further improving the efficiency in an economy where new technologies are lacking. The study will estimate the technical efficiency of the farms in the regions of Jammu and Kashmir using the conventional DEA method and bootstrapping procedures. The allocative efficiency and cost efficiency of the farms are also estimated which will help to identify the variables at farm levels that influence farmers' technical efficiency and determine the opportunities for increasing the farm yield. Keeping in view the importance of fish farming in the economy and life of the people of Jammu and Kashmir in general, and exotic carp in particular, it is of immense importance to understand the efficiencies in carp production in the state to develop a sound plan for the development of fisheries/aquaculture in the state of Jammu and Kashmir.

Data envelopment analysis (DEA) is a non-parametric production frontier approach that measures the effectiveness of a sample farm considering the inputs used in the culture or maximum amount of output produced with its existing resources. Under the output-based approach, performance is judged by the capacity to create the greatest yields achievable from a given set of inputs' technical effectiveness or to maximize income given yield costs and input amounts (economic efficiency) [7]. Given input prices and outputs, performance is assessed in terms of the greatest possible reductions in input quantities (cost efficiency). This method is advantageous since it does not impose a functional form a priori and allows for a variety of output technologies [8]. The proportion of economic and technical efficiencies determines the degree of allocative efficiency in each case. Allocative efficiency reflects the ability of the firm to produce an optimum combination of different outputs, while under the input-based approach, it reflects the ability to use inputs in an optimum proportion. The foremost feature of DEA is that it produces a single input–output index to characterize the effectiveness of a firm or decision-making unit DMU, creating multiple outputs from a set of inputs [9].

The paper presents statistical inferences about the technical efficiency estimated using data envelopment analysis (DEA) and the bootstrap method. The findings of the bootstrap approaches are also compared to those of the conventional DEA approach. The case of exotic freshwater carp polyculture in northern India (Jammu and Kashmir) is used for the analysis. The objectives of the study are: (1) To compare the technical efficiency of the farms using the conventional DEA method and the single bootstrap method. (2) Estimation of allocation and cost efficiencies of the farms to help identify the variables that could assist in increasing the farm yield. (3) To regress the estimated efficiency scores with the socio-economic variable using tobit regression to identify factors that are significant in explaining differences in levels of efficiency between exotic carp farms.

## 2. Review of Literature

DEA is a non-parametric approach that measures the efficiency (technical, allocative, and cost) of a set of decision-making units with multiple inputs and outputs and by using mathematical programming. Charnes [10] initially proposed the model of DEA and later it was developed and applied in wide range of fields. Wang [11] in his study evaluating the performance of the fishing industry in Vietnam used a DEA model and provided a general overview via technical efficiency and total factor productivity. Zhu [12] in his study inspecting 204 Iceland companies used a DEA model and revealed that a large percentage of companies were performing inefficiently during the production process. Anh [13] studied the technical inefficiency of Vietnamese pangasius farming using data envelopment analysis and revealed that inadequate use of capital assets (42%) and improper methods to achieve higher yields (30%) are the main challenges for enhancing the efficiency of the fish culture. Sangun [14] aimed to analyze the economic efficiency of the small-scale coastal fishing activities operating on the Eastern Mediterranean coast of Turkey by using DEA. The study revealed that the average economic efficiency of these activities under variable returns to scale is 62%. Long et al. [15] applied DEA bootstrap methods to analyze the efficiency in Vietnam's intensive white-leg shrimp farming. The findings expounded the technical efficiency of 0.69 and the significant positive influence on the efficiency is from large farm sizes and access of formal credit.

For many years, conventional DEA methods have been used for estimation of technical efficiency in aquaculture. However, when using the DEA approach, making statistical judgments about technical efficiency is challenging for the following reasons. Primarily, all technical efficiency estimates are based on a single sample. Despite the fact that the traditional DEA technique is deterministic, the efficiency is still calculated in terms of an estimated rather than realistic frontier. The efficiency scores obtained from a finite sample are sensitive to the calculated frontier's sampling variation [8]. Secondly, according to the DEA assumption [16], calculated DEA in a production set is a smaller representative subset of the actual considered population, resulting in optimistic estimated technical efficiency measurements. As a result, when attempting to draw general policy implications in aquaculture from a specific group of related but independent studies, caution is advised. To address these issues, the authors of [16,17] integrated the smoothed bootstrap technique into the DEA framework to present the nonparametric frontier model's statistical base. To incorporate the bootstrap into the DEA framework, assume that the data generation process (DGP) generated the initial sample data and that the DGP can be stimulated by using a "new" or "pseudo" data set obtained from the original data set. An empirical distribution of these bootstrap values can be obtained by repeating this method several times. This produces a Monte Carlo approximation of the sampling distribution, making inference methods easier. The single bootstrap methods, primarily based on precise statistical models, provide a reliable estimate of the technical efficiency, standard error, hypothesis testing, confidence interval, and the production frontier [15].

To analyze the technical efficiency in aquaculture production, returns to scale is an important issue and as per [18], we only consider the constant returns to scale DEA model when the farms are operating at optimal levels. Many obstacles, such as farmer socioeconomic restrictions, imperfect competition, financial limits, etc., obstruct optimal farm operations [19,20]. Therefore, in a developing country such as India, the variable returns to scale (VRS) DEA model is assumed in the production technology, especially in case aquaculture studies [21–24]. In addition, Simar and Wilson [25] presented returns-to-scale tests based on the single bootstrap approach to improve precision in the assessment of the border of production possibilities set in the nonparametric analysis [15,26,27].

## 3. Materials and Methods

### 3.1. Research Flow

The author explained the variables and utilized the DEA model for measuring technical, allocative, and cost efficiencies of exotic fish polyculture in Jammu and Kashmir, India. The research flow is explained in Figure 1 as follows.

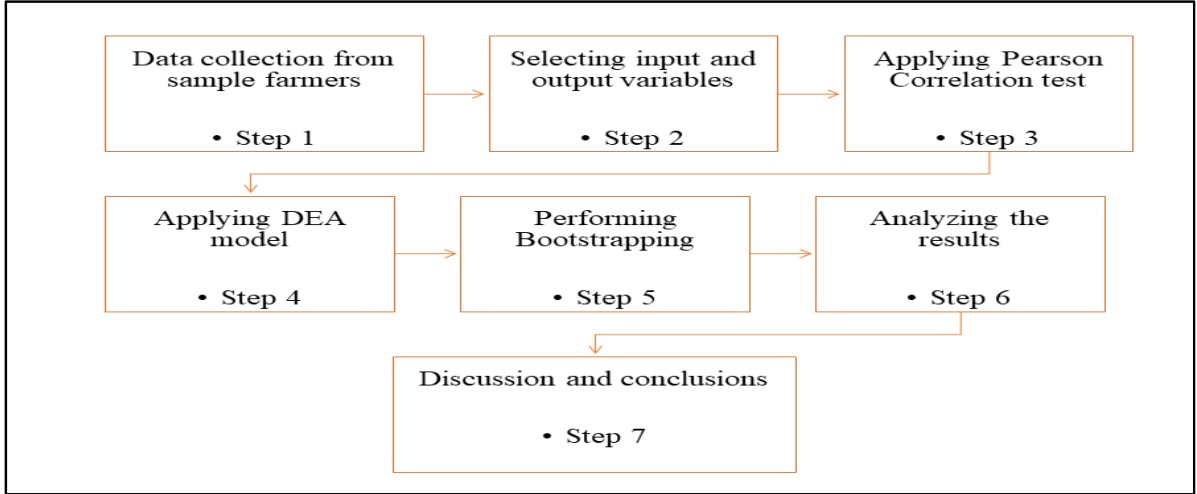

**Figure 1.** Steps to follow in DEA analysis.

Step 1: Data collection from sample farmers.

A pre-tested questionnaire especially designed for the study was used to collect the information from the sample farms practicing polyculture of exotic carp in the region. By using multistage stratified simple random sampling and the snowball technique, the farmers were identified and interviewed.

Step 2: Selecting input and output variables.

In order to run the DEA model, this step is very crucial and only those variables that could possibly contribute to enhancing the efficiency were selected. In this model, the monetary value of the total fish produced was used as output and inputs used were number of seeds stocked, amount of feed used (rice bran and mustard oil cake), quantity of lime used, and labor. The correlation between the variables significantly contributes to precision and accurate representation of the results.

Step 3: Applying the Pearson correlation test.

The Pearson correlation test was applied to understand the statistical relationship between the inputs and output variables. The test explains the linear correlation between the two set of data used in the model. If there is a positive correlation coefficient between the inputs and output, it means accurate selection of variables and, further, that the data can be used for DEA modeling.

Step 4: Applying the DEA model.

Data envelopment analysis (DEA) is a non-parametric production frontier method that assesses the efficiency of a sample farm based on the inputs used in the culture or the maximum amount of output achieved with the farm's current resources. The model was applied to effectively estimate the efficiency of exotic fish polyculture farming in the northern part of India.

Step 5: Performing bootstrapping.

Bootstrapping is a statistical technique used for generating numerous simulated samples from a single dataset. This procedure allowed us to compute standard errors, create confidence intervals, and undertake hypothesis testing.

Step 6: Analyzing the results.

Data envelopment analysis (DEA) is used for the estimation of production frontiers and in this paper the author used it to empirically measure productive efficiency of sample exotic fish farms. After applying DEA, the researcher analyzed the data (inputs and output variables) and summarized and presented the results in graphical and tabular forms.

Step 7: Discussion and conclusion.

After successfully analyzing the results, author summarized the research findings and highlighted the key significant contributions in the studied area.

### 3.2. Pearson's Correlation Coefficient

The presence of a significant relationship between the input and output variables is essential before utilizing the DEA model. The prior condition for using Pearson's correlation test is the linear relationship between the variables that can be either positive or negative as long as it is significant. The method is based on covariance and is considered the best method for measuring the association between the variables. A closer relationship between the variables is denoted by a higher correlation coefficient and vice versa. The range of the correlation coefficient is always between −1 and +1 and is presented in the equation below.

$$r_{xy} = \frac{\sum_{i=1}^{n}(x_i - \bar{x})(y_i - \bar{y})}{\sqrt{\sum_{i=1}^{n}(x_i - \bar{x})^2 \ \sum_{i=1}^{n}(y_i - \bar{y})^2}} \tag{1}$$

where *n* is the size of the sample, $x_i$ and $y_i$ denote the individual sample points related to *i*.

### 3.3. Data Description and Sampling Procedure

The study was carried out in August 2017 in the state of Jammu and Kashmir that is in the northernmost region of the Indian subcontinent. The data was collected from sample farmers culturing exotic fish polyculture (grass carp, common carp, and silver carp). For sampling, Jammu district from the Jammu region and two districts *viz.* Ganderbal and Budgam districts from the Kashmir region were selected because of the higher prevalence of exotic carp polyculture in these regions. A total of 160 exotic carp households were selected out of which 80 were from Jammu district (Jammu region) and 40 each from Ganderbal and Budgam districts (Kashmir region).

The primary data was collected from farmers using multistage stratified simple random sampling and the snowball technique and was collected with a personal interview method with the help of a pre-tested questionnaire especially designed for the study. Secondary data, on the other hand, was gathered from relevant journals and books. The questionnaire elicited information on the average farm size, stocking density, feeding rates, days of culture, equipment used, average size of carp harvested, and production input quantities and costs. The farmers were asked to provide the information for the previous production year and the response rates for the survey were 98.70%.

### 3.4. Analytical Technique

In this study, a two-staged DEA-tobit model [28] was employed in order to first estimate the technical efficiency scores and later regress the estimated technical efficiency scores with the socio-economic variable using the tobit regression technique in STATA. This analytical procedure yields better results than eith single staged or double staged SFA that assumes Cobb–Douglas production from Banker et al. [29]. Data envelopment analysis (DEA) is an alternative non-parametric method of measuring efficiency that uses numerical programming instead of regression. Here, one circumvents the issue of specifying an explicit form of the production function and makes only a minimum number of assumptions about the underlying technology. Farrell [30] defined a linear programming model to measure the technical efficiency of a firm with reference to a benchmark technology characterized by steady returns to scale. This effectiveness measure corresponds to the

coefficient of resource utilization defined by Debreu [31]. To address the constraints of efficiency estimation for decision making units (DMUs) having numerous inputs and outputs without market prices, the authors of [10] introduced the method of data envelopment analysis (DEA).

Suppose that N farms used k inputs and further produced m outputs for each product. The vectors xi and yi, respectively, represent the ith farm. In this non-parametric approach, the yield (output) is represented in the form of a ratio, which is the reduced form of numerous inputs and outputs. Therefore, UiYi/ViXi is a measure of the ratio of all outputs over all inputs (where U is an M × 1 vector of output weight and V is a K × 1 vector of input weight) which is ultimately the measure of efficiency. Hence, for farm 1, the following mathematical programming model must be solved.

$$
\begin{aligned}
&\text{Max u. v (U, Y, /V, X,),}\\
&\text{Subject to:}\\
&\text{U, Y, /V, X, = 1, I = 1, 2, 3 \ldots N (U and V are variable weights)}\\
&\text{U, V = 0}
\end{aligned}
\tag{2}
$$

Here, xi and yi are the vectors of the ith farm. As the ratio is maximized, it would be constrained to be no greater than one; thus, all firms in the sample are forced to be on or below the frontier.

However, to avoid the problem of an infinite number of solutions (that is, if U″, V″), a solution variable then [$\alpha$U″, $\alpha$V *] is another solution, etc., a constraint Vx, = 1 is imposed on the fractional programming model in Equation (2), and this is formulated as the following linear program:

$$
\begin{aligned}
&\text{Min}_{\theta\,\lambda}\ \theta\\
&\text{Subject to}\\
&-Y + Y_\lambda \geq 0\\
&\theta = \text{free, } \lambda \geq 0
\end{aligned}
\tag{3}
$$

where $\theta$ is scalar and is an N × 1 vector of constraints.

The above exposition is based on input orientation and assumed constant returns to scale [CRS]. However, Banker et al. [29] proposed a variable returns to scale (VRS) model to be used in a situation where the industries are not perfectly competitive. Extending the CRS linear programming and adding the convexity constraints resulted in the following:
N 1′ $\lambda$ = 1 to Equation (3) to provide

$$
\begin{aligned}
&\text{Min, } \theta\\
&\text{Subject to:}\\
&-Y_i + Y_\lambda \geq 0\\
&\text{N 1′ } \lambda - 1\\
&\lambda \geq 0
\end{aligned}
$$

where N1 is an N × 1 vector of ones

This approach based on the work of Farrell [30] and Fare et al. [7] has since been improved upon and extended by [32,33]. Charnes [9] also developed the fractional linear programming method of DEA, which compares inefficient firms with the 'best practice' ones within the same group. The current level of efficiency is not ideal, it may be progressed upon and can be raised as suggested by Ajibefun [34] in his study on small-scale food crop farmers in Ondo state, Nigeria.

In any case, the DEA approach suffers from criticisms that it takes no account of the conceivable impact of measurement errors and other noise data that are common in fisheries, since all observed deviations from the evaluated frontier are expected to be the result of technical inefficiency [18].

### 3.5. The Bootstrap Proposed by Simar and Wilson

Simar and Wilson [16,17] proposed the single bootstrap method for input-oriented variable returns to scale. The DEA on the basis of the smoothed bootstrap process is shown below.

Step 1: Compute the estimate of technical efficiency, $\hat{\theta}_j$ the jth farm as in Equation (1).

Step 2: Use bootstrap via smooth sampling from $\hat{\theta}_1 \dots , \hat{\theta}_N$ to obtain a bootstrap replica $\theta_N^*$: this is completed as follows:

Bootstrap, sample with replacement from $\hat{\theta}_1, \hat{\theta}_N$ and call the results $\beta_1, \dots \beta_N$.

Simulate standard normal independent random variables $\varepsilon_1, \varepsilon_N$. Calculate $\hat{\theta}_j =$
$$\begin{cases} \beta j + h\varepsilon j \ if \ \beta j + h\varepsilon j \ \leq 1 \\ 2 - \beta j - h\varepsilon j \ otherwise \end{cases}$$ . Note that $\hat{\theta}_j \leq 1$ and h is the band width.

Adjust $\hat{\theta}_j$ to obtain parameters with asymptotically correct variance, and then estimate the variance $\hat{\sigma}^2 = \frac{1}{N} \sum_{j=1}^{N} (\hat{\theta}_j - \hat{\theta})$ and calculate $\theta_j^* = \frac{1}{\sqrt{1 + \frac{h^2}{\hat{\sigma}^2}}} (\hat{\theta}_j - \hat{\beta})$ where $\hat{\beta} = \frac{1}{N} \sum_{j=1}^{N} \beta j$.

Step 3: For $j = 1, N$, a pseudo data set of $(x_{j,b}^*, \ y_{j,b}^*)$.

where $x_{j,b}^* = \frac{\hat{\theta}_j}{\theta_j^*} x_j$ and, $y_{j,b}^* = y_j$. Calculate the new DEA score $\hat{\theta}_j^*$ for each fish farm from Equation (1) by taking the pseudo data as reference.

Step 4: Repeat steps (1) to (3) for B times to yield a new DEA technical efficiency score $\hat{\theta}_j^*$ for $j = 1, N$, Therefore the bias corrected estimator of $\hat{\theta}_j$ can be computed as $\hat{\hat{\theta}}_j = B^{-1} \sum_{b=1}^{B} \hat{\theta}_j^*$.

Step 5: The confidence interval of a $(1-a)$ level for the technical efficiency can be established by finding value $a_a, b_a$ such that $Pr(-a_a \leq \hat{\theta}_j - \hat{\theta} \leq -b_a) = (1-a)$. Since we do not know the distribution of $(\hat{\theta}_j - \hat{\theta})$, we can use the bootstrap values to find $\hat{a_a}, \hat{b_a}$, such that $Pr(-\hat{a_a} \leq \hat{\theta}_j^* - \hat{\hat{\theta}}_j \leq \hat{b_a}) = (1-a)$. Therefore, the estimated confidence level of $(1 - a)$ for technical efficiency of the jth exotic farm is $\hat{\theta}_j + \hat{b_a} \leq \theta_j \leq \hat{\theta}_j + \hat{a_a}$.

### 3.6. Tobit Regression Explaining Determinants of Efficiency

Efficiency scores obtained from the DEA at the first stage were regressed on farmers' characteristics at the second stage using tobit regression. The efficiency scores regressed with farmers' characteristics vary from zero to unity and hence are appropriate.

The tobit regression takes the following form:

$$EFF = \beta_0 + \beta_1 \ EDU + \beta_2 \ EXP + \beta_3 \ AGE + \beta_4 \ FAM + e$$

where EFF represents the efficiency scores (ranging from 0 to 1) of the farms obtained from DEA. EDU is the total years of education, EXP is the total years of exotic fish culture experience, AGE is the age of the farmers expressed in years, and FAM is the total number of family members.

The intercept captured the location, Jammu and Kashmir. Maximum likelihood estimate was used to estimate the tobit model. STATA was used to run the tobit regression. SPSS was used to determine the relationship between variables

## 4. Results and Discussion

### 4.1. Socio-Economic Characteristics of Farmers

The general characteristics of farmers in Jammu and Kashmir such as gender, age, family type, family size, education level, and occupation are presented in Table 1. A total of 160 farmers, 80 each from Jammu region and Kashmir region were interviewed and it is observed that majority of farmers in the Jammu region as well as in the Kashmir region were males. Out of the total farmers, only 10% and 17.5% were females in Jammu and

Kashmir, respectively. The sample fish farmers were grouped on the basis of their age into three categories (<45 years, 46–55 years, and greater than 55 years) for convenience in the analysis. The majority of fish farmers in Jammu (60%) were in the age group of less than 45 years of age followed by the age group of 46–55 years of age (32.50%) and greater than 55 years of age (7.5%). Similarly, in Kashmir region the majority (47.5%) of farmers were in the age group of 46–55 years followed by less than 45 years of age (37.5%) and greater than 55 years of age (15%). This also shows that the farmers in Jammu were more dynamic young people who started their business at a young age, but the majority of farmers in Kashmir were in their middle years and had more experience.

**Table 1.** Characteristics of sample farm households.

|  | | Jammu | | Kashmir | |
|---|---|---|---|---|---|
|  | Category | No. | Share (%) | No. | Share (%) |
| **Sample size** | | 80 | | 80 | |
| **Gender** | Male | 72 | 90 | 66 | 82.5 |
|  | Female | 8 | 10 | 14 | 17.5 |
| **Age** | <45 years | 30 | 37.5 | 48 | 60 |
|  | 46–55 years | 38 | 47.5 | 26 | 32.5 |
|  | ≥56 years | 12 | 15 | 6 | 7.5 |
| **Family Type** | Joint | 50 | 62.5 | 32 | 40 |
|  | Nuclear | 30 | 37.5 | 48 | 60 |
| **Family Size** | 2–4 members | 24 | 30 | 38 | 45 |
|  | 5–7 members | 22 | 27.5 | 26 | 32.5 |
|  | >7 members | 34 | 42.5 | 18 | 22.5 |
| **Education Level** | Illiterate | 2 | 2.5 | 6 | 7.5 |
|  | Primary | 34 | 42.5 | 20 | 25 |
|  | Secondary | 30 | 37.5 | 36 | 40 |
|  | Higher secondary | 14 | 17.5 | 16 | 20 |
|  | Graduate | 0 | 0 | 4 | 5 |
|  | PG | 0 | 0 | 2 | 2.5 |
| **Occupation** | Agriculture | 60 | 75 | 66 | 82.5 |
|  | Business | 6 | 7.5 | 6 | 7.5 |
|  | Govt. Job | 14 | 17.5 | 8 | 10 |

When studying the family type of the farmers, Jammu region was the highest (62.5%) in the case of joint families compared to the Kashmir farmers (60%). A joint family in a farming enterprise plays a vital role in their benefit and cost ratio: the larger the family, the more family members involved in the farming practice, which reduces the cost of hired labor. The education profile of farmers in Jammu revealed that most farmers had a primary level of education (42.50%) followed by secondary level (37.5%) and higher secondary (17.50%). Likewise for the sample of fish farmers in Kashmir, the majority of the farmers had secondary education (40%) followed by primary (25%) and higher secondary (20%).

*4.2. Fixed Capital Investment Pattern*

The fixed capital investment pattern on sample carp farms was investigated in order to better understand the farmer's revenue generating potential. The investment patterns for the regions are presented in Table 2. The culture of exotic carp in the state is practiced in ponds constructed under the RKVY (Rashtriya Krishi Vikas Yojana) scheme launched by

Government of India to promote fish culture in the state. All the carp farms studied were below the area of 1 hectare (0.15 ha) with 0.1 ha of water spread area.

**Table 2.** Fixed capital investment pattern on sample carp farms (USD).

| Particulars | Jammu | | | Kashmir | | |
|---|---|---|---|---|---|---|
| | **Per Farm** | **Per Hectare** | **Share (%)** | **Per Farm** | **Per Hectare** | **Share (%)** |
| **Pond construction** | USD 670.60 | USD 6773.41 | 50.66 | USD 1316.93 | USD 13,302.95 | 61.10 |
| **Inlet/outlet** | USD 94.45 | USD 953.94 | 7.13 | USD 182.86 | USD 1846.88 | 8.48 |
| **Farm building** | USD 96.92 | USD 979.16 | 7.32 | USD 149.69 | USD 1511.86 | 6.94 |
| **Power connection and lighting** | USD 62.10 | USD 626.87 | 4.69 | USD 40.71 | USD 411.51 | 1.89 |
| **Nets** | USD 210.01 | USD 2121.73 | 15.87 | USD 267.85 | USD 2705.96 | 12.43 |
| **Electric motor** | USD 189.58 | USD 1915.15 | 14.32 | USD 197.40 | USD 1993.83 | 9.16 |
| **Total** | USD 1323.65 | USD 13,370.25 | 100.00 | USD 2155.59 | USD 21,772.45 | 100.00 |

The fixed capital investments were estimated on both per farms as well as on per hectare basis. It was observed that the fish farms in Kashmir valley depicted higher investments in comparison to that in the Jammu region, the major reason being the higher cost of pond construction attributed probably to the high cost of available human labor. The highest share in investment was pond construction that accounted for 50.66% for the farmers in the Jammu region and 61.10% for sample farms in Kashmir. Griffin et al. [35] also found that pond construction accounts for 45–49% of the total investment, whereas land and water charges account for 25–28% of the total cost. The other investments on the farms were inlet–outlet, farm building, power connection, lighting, and electric motors.

*4.3. Summary of Descriptive Statistics*

The major inputs used in the production of exotic carp (q/ha) in the sample farms of the state include rice bran (q/ha), mustard oil cake (q/ha), lime (q/ha), seed (no./ha), and labor (days/ha). In order to feed the exotic carp, the farmers made a mixture of rice bran and mustard oil cake in the ratio of 1:1 and then fed them twice daily. The statistical analysis of the variables is presented in Table 3 and revealed a large variation in the input use among the farmers in both regions. The highest variation was seen in the use of lime where some of the farmers used around 6 times less than the maximum followed by labor, which was about 3 times less than the maximum. The possible reasons for the variation could be due to inefficiency in available resource use.

**Table 3.** Statistics of exotic carp outputs and inputs used in the state.

| | Jammu | | | | Kashmir | | | |
|---|---|---|---|---|---|---|---|---|
| **Output/Input Variables** | **Min** | **Max** | **Mean** | **St. dev** | **Min** | **Max** | **Mean** | **St. dev** |
| | | | **Output** | | | | | |
| Fish yield (q/ha) * | 119.60 | 174.75 | 133.74 | 16.57 | 82.63 | 137.78 | 109.91 | 9.49 |
| | | | **Inputs** | | | | | |
| Rice bran (q/ha) * | 63.26 | 108.72 | 75.53 | 13.94 | 60.29 | 128.49 | 70.62 | 12.48 |
| Mustard oil cake (MOC) (q/ha) * | 42.90 | 117.82 | 73.14 | 16.43 | 61.28 | 97.85 | 70.81 | 10.32 |
| Lime (q/ha) * | 1.58 | 8.90 | 6.11 | 2.05 | 2.09 | 10.23 | 5.77 | 1.89 |
| Seed stocked (No./Ha) | 19,372.64 | 52,385.20 | 37,687.69 | 10,118.49 | 18,186.56 | 39,536.00 | 20,852.83 | 3249.45 |
| Labor (days/ha) | 138.38 | 494.20 | 212.66 | 77.70 | 98.84 | 336.06 | 167.79 | 46.17 |

* (q/ha) is the total weight taken in quintals per hectare of area.

*4.4. Correlation Coefficient*

The association between the input and output variables was studied using Pearson's correlation coefficient and is presented in Table 4. It is evident from the results that the input and output variables are positively correlated and significant at 0.01 and 0.05 levels. Hence, the data set is fit and suitable for evaluating and analyzing the performance of exotic carp farms using the DEA method

**Table 4.** Correlation coefficient between variables.

| | | Jammu | | | | | | Kashmir | | | | | |
|---|---|---|---|---|---|---|---|---|---|---|---|---|---|
| | | RB | MOC | Lime | Lab | Seed | Output | RB | MOC | Lime | Lab | Seed | Output |
| **RB** | Pearson Correlation | 1 | 0.981 ** | 0.469 ** | 0.520 ** | 0.707 ** | 0.368 * | 1 | 0.981 ** | 0.469 ** | 0.520 ** | 0.707 ** | 0.368 * |
| | *p*-value | | 0.000 | 0.002 | 0.001 | 0.000 | 0.020 | | 0.000 | 0.002 | 0.001 | 0.000 | 0.020 |
| | Sample size | 80 | 80 | 80 | 80 | 80 | 80 | 80 | 80 | 80 | 80 | 80 | 80 |
| **MOC** | Pearson Correlation | 0.981 ** | 1 | 0.470 ** | 0.532 ** | 0.702 ** | 0.278 * | 0.981 ** | 1 | 0.470 ** | 0.532 ** | 0.702 ** | 0.277 * |
| | *p*-value | 0.000 | | 0.002 | 0.000 | 0.000 | 0.013 | 0.000 | | 0.002 | 0.000 | 0.000 | 0.017 |
| | N | 80 | 80 | 80 | 80 | 80 | 80 | 80 | 80 | 80 | 80 | 80 | 80 |
| **lime** | Pearson Correlation | 0.469 ** | 0.470 ** | 1 | 0.217 ** | 0.283 * | 0.188 * | 0.80 ** | 0.470 ** | 1 | 0.216 * | 0.283 * | 0.188 * |
| | *p*-value | 0.002 | 0.002 | | 0.001 | 0.026 | 0.044 | 0.002 | 0.002 | | 0.017 | 0.016 | 0.244 |
| | N | 80 | 80 | 80 | 80 | 80 | 80 | 80 | 80 | 80 | 80 | 80 | 80 |
| **lab** | Pearson Correlation | 0.520 ** | 0.532 ** | 0.217 * | 1 | 0.430 ** | 0.320 * | 0.520 ** | 0.532 ** | 0.216 * | 1 | 0.430 ** | 0.320 * |
| | *p*-value | 0.001 | 0.000 | 0.013 | | 0.006 | 0.044 | 0.001 | 0.000 | 0.017 | | 0.006 | 0.044 |
| | N | 80 | 80 | 80 | 80 | 80 | 80 | 80 | 80 | 80 | 80 | 80 | 80 |
| **seed** | Pearson Correlation | 0.707 ** | 0.702 ** | 0.283 * | 0.430 ** | 1 | 0.364 * | 0.707 ** | 0.702 ** | 0.283 * | 0.430 ** | 1 | 0.364 * |
| | *p*-value | 0.000 | 0.000 | 0.016 | 0.006 | | 0.021 | 0.000 | 0.000 | 0.025 | 0.006 | | 0.021 |
| | N | 80 | 80 | 80 | 80 | 80 | 80 | 80 | 80 | 80 | 80 | 80 | 80 |
| **output** | Pearson Correlation | 0.368 * | 0.278 * | 0.188 * | 0.320 * | 0.364 * | 1 | 0.368 * | 0.277 * | 0.188 * | 0.320 * | 0.364 * | 1 |
| | *p*-value | 0.020 | 0.033 | 0.024 | 0.044 | 0.021 | | 0.020 | 0.025 | 0.014 | 0.044 | 0.021 | |
| | N | 80 | 80 | 80 | 80 | 80 | 80 | 80 | 80 | 80 | 80 | 80 | 80 |

Note **. Correlation is significant at the 0.01 level *. Correlation is significant at the 0.05 level. RB denotes rice bran and MOC denotes mustard oil cake.

### 4.5. Technical Efficiency Estimation

In this study, data envelopment analysis (DEA) was used to evaluate the efficiency and performance of exotic carp polyculture in India. Table 5 illustrates estimated levels of efficiency using conventional DEA and single bootstrapping approaches. In the region of Jammu, the average technical efficiency using the conventional DEA model for all the farms is 0.9771. This implies that the sample exotic carp farmers can reduce their input by 2.29% without changing their level of output. Similarly, for Kashmir, the average technical efficiency using the conventional DEA method is 0.9741 indicating the reduction of their input by 2.59%.

**Table 5.** Conventional DEA and bootstrap TE estimates.

|  | **Jammu** | | | | **Kashmir** | | | |
|---|---|---|---|---|---|---|---|---|
|  | **Min** | **Maximum** | **Mean** | **SD** | **Min** | **Maximum** | **Mean** | *SD* |
| **Conventional TE** | 0.8129 | 1 | 0.9771 | 0.0706 | 0.8012 | 1 | 0.9741 | 0.0519 |
| **Bootstrap-corrected TE** | 0.6667 | 1 | 0.9744 | 0.0706 | 0.7286 | 1 | 0.9727 | 0.0517 |
| **Higher bound single** | 0.813 | 1.18 | 0.9771 | 0.0708 | 0.8012 | 1.27 | 0.9735 | 0.0525 |
| **Lower bound single** | 0.6895 | 0.967 | 0.9728 | 0.0707 | 0.7317 | 0.993 | 0.9213 | 0.0523 |

From these efficiency estimates, we applied the single bootstrapping method for correcting the bias of technical efficiency as done by [16,17]. The average bias corrected technical efficiency (TE) for single bootstrap is 0.944 for Jammu region and 0.9727 for Kashmir. Additionally, the lower and upper boundaries for the 95% confidence interval for bias-corrected technical efficiency for the single bootstrap method is 0.6895 and 0.813 for sample farms in Jammu region, indicating that an increased technical efficiency might save the average farm between 18% and 31% of its input. At the same time, the farms in Kashmir region can save up to 20–27% of their inputs. The width of the 95% confidence interval of technical efficiency for the single bootstrap is 0.1235 for Jammu and 0.067 for Kashmir, respectively. The results revealed that farmers in both the regions are efficiently utilizing their available resources with a small average technical inefficiency of 3%.

### 4.6. Allocative and Cost Efficiencies

Economic efficiency is a combination of technical and allocative efficiencies. The estimation of technical efficiency is of vital importance in order to understand how farmers can increase the production without paying for the extra costs and with existing production technology as suggested by [36]. Efficiencies were estimated in terms of allocative and cost efficiencies for each region using the Data Envelopment Analysis Program (DEAP) version 2.1(Tim Coelli, Armidale, Australia) by [18]. The average efficiency scores of the sample farms are reported in Table 6 and revealed that farmers in Jammu region were more allocative efficient (0.92) and less cost efficient (0.75) and Kashmir revealed a similar pattern of higher allocative efficiency than cost efficiency. The mean allocative efficiency clearly indicates that the farmers were able to produce the output (carp production) using the combination of inputs (feed, seed, fertilizers, etc.) corresponding to the minimum cost of production.

The majority of the farmers in Jammu region had allocative efficiencies between 0.901 and 1 (65%) followed by the efficiency between 0.801 and 0.9 (25%) and 0.701–0.8 (10%). At the same time, farmers of Kashmir showed the highest allocative efficiency between 0.801 and 0.9 (40%) followed by 0.701–0.8 (30%) and 0.901–1 (30%). This rather high degree of efficiency in both the regions recommends that exceptionally small marketable yield is sacrificed to resource waste. The analysis of allocative efficiency depends on assumptions made about a farmer's behavior. Farrell [30] assumed that farmers allocate resources on the basis of cost minimization to obtain a given level of output. Allocative inefficiency is a famer's inability to equate the ratio of marginal products to the ratio of their respective prices. Kopp and Diewert [37] characterized allocative inefficiency as the failure to compare

the marginal value product of inputs to their prices. In the DEA model, however, the behavioral assumption is more sensitive as the allocative effectiveness is the proportion by which the costs of the levels of inputs on a fish farm can be decreased without any loss in output.

Frequency of allocative efficiency was found highest in the interval of 0.901–1 with 65% of farmers falling in the range. The farmers of Jammu region were more allocative efficient than cost efficient with an average of 0.92, indicating that the farmers were producing the fish with optimum combination of inputs. The mean allocative and cost efficiencies are presented in Figure 2 depicting the percentage of farmers falling in each efficiency group.

**Table 6.** Distribution of allocative and cost efficiencies of sample farms in the region.

| | Jammu | | | | Kashmir | | | |
|---|---|---|---|---|---|---|---|---|
| | *Allocative Efficiency* | | *Cost Efficiency* | | *Allocative Efficiency* | | *Cost Efficiency* | |
| *Efficiency level* | Freq. | % | Freq. | % | Freq. | % | Freq. | % |
| *0.501–0.6* | 0 | 0 | 8 | 10 | 0 | 0 | 4 | 5 |
| *0.601–0.7* | 0 | 0 | 16 | 20 | 0 | 0 | 16 | 20 |
| *0.701–0.8* | 8 | 10 | 24 | 30 | 24 | 30 | 44 | 55 |
| *0.801–0.9* | 20 | 25 | 18 | 22.5 | 32 | 40 | 10 | 12.5 |
| *0.901–1.00* | 52 | 65 | 14 | 17.5 | 24 | 30 | 6 | 7.5 |
| *Total* | 80 | 100 | 80 | 100 | 80 | 100 | 80 | 100 |
| *Mean efficiency* | 0.92 | | 0.75 | | 0.84 | | 0.74 | |
| *Median* | 0.94 | | 0.72 | | 0.83 | | 0.73 | |
| *Mode* | 0.969 | | 1 | | 0.74 | | 0.63 | |
| *Minimum* | 0.716 | | 0.475 | | 0.73 | | 0.57 | |
| *Maximum* | 1 | | 1 | | 1 | | 1 | |
| *Std. Deviation* | 0.067 | | - | | 0.071 | | 0.088 | |

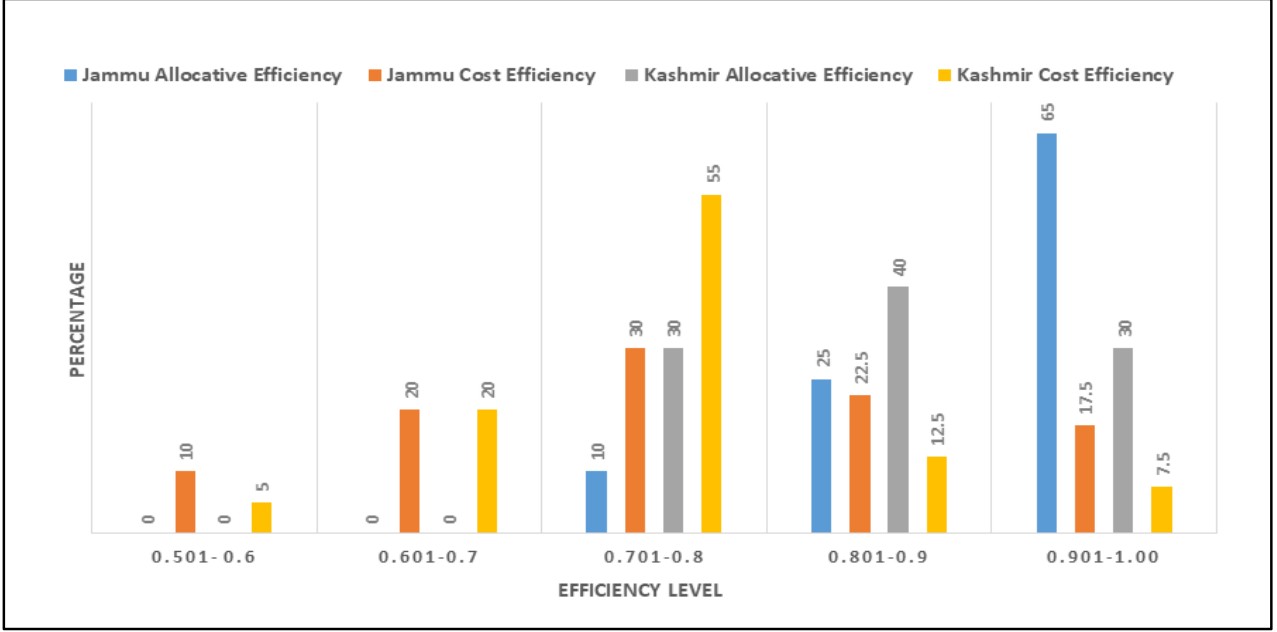

**Figure 2.** Distribution of allocative and cost efficiency measures.

*4.7. Technical Inefficiency Analysis*

The study used respective efficiencies as the dependent variable and thus those variables with a positive–negative coefficient sign depicted the positive–negative impact on efficiency measures. The results of the tobit regression for Jammu and Kashmir are presented in Table 7. The coefficient of family number was found to be negative and statistically significant both in the Jammu as well as in the Kashmir region and this negative relationship implies that as the family size increases, farmer's technical effectiveness decreases. This indicates that as the family size increases, more family members indulge in farming practices which ultimately reduces the dependence on hired skilled labor increasing the inefficiency. Contrary to this, education, experience, and age turned out to very significant variables explaining the economic efficiency of the exotic carp farms in Jammu and Kashmir. Yusuf and Malomo [38] obtained similar results where the technical efficiency of the poultry farmers of Ogun state was positively affected by years of experience and education. The implication is that the farmers with more farming experience and who are older are more technically efficient which may be due to the reason that with experience the farmers are better able to cope with the limitations that hinder the productivity and reduces the errors in farming. Mohan et al. [39] also could not relate any significance of age in determining the inefficiency in India, Vietnam, Thailand, and China. Analyses suggest that small-scale farms (<1 ha) as selected in this study are easily managed, require less investment, and fewer risk factors are involved. Sharma et al. [40] and Yin et al. [41] obtained similar results where the technical efficiency of the farms decreases as the size increases.

**Table 7.** Tobit estimates of the variables of efficiency function in Jammu and Kashmir.

| | Jammu | | | | | | Kashmir | | | | | |
|---|---|---|---|---|---|---|---|---|---|---|---|---|
| | Technical Efficiency | | Allocative Efficiency | | Cost Efficiency | | Technical Efficiency | | Allocative Efficiency | | Cost Efficiency | |
| | Coeff | t-Ratio | Coeff | t-Ratio | Coeff | t-Ratio | Coeff | t-Ratio | Coeff | t-Ratio | Coeff | t-Ratio |
| Constant | 0.529 | 4.720 * | 0.909 | 9.500 * | 0.479 | 3.910 * | 0.871 | 11.260 | 0.880 | 8.170 * | 0.774 | 7.050 ** |
| Education | 0.008 | 1.480 ** | 0.004 | 0.810 | 0.010 | 1.760 ** | 0.002 | 0.540 * | 0.002 | 0.430 | 0.003 | 0.710 |
| Experience | 0.010 | 1.720 | 0.002 | 0.430 * | 0.012 | 1.850 ** | 0.005 | 1.660 * | −0.004 | −1.100 | 0.000 | −0.010 |
| Age | 0.005 | 2.430 * | −0.001 | −0.640 | 0.003 | 1.530 * | 0.002 | 1.670 | 0.000 | −0.080 | 0.001 | 0.850 |
| Family number | −0.014 | −1.580 * | 0.001 | 0.090* | −0.012 | −1.250 | −0.018 | −4.320 | −0.005 | −0.900 | −0.020 | −3.430 ** |
| Log likelihood | 44.20 | | 50.32 | | 40.67 | | 63.63 | | 50.37 | | 49.63 | |

Note: Single (*) and double (**), denotes, respectively, significance at the 10%, and 5% levels, TE denotes technical efficiency, AE denotes allocative efficiency, and CE denotes cost efficiency.

## 5. Conclusions

The primary goal of this study was to determine the input–output specific technical and scale efficiency of Indian exotic carp production in order to identify possible areas for improvement and to analyze the impact of farmer and farm attributes on these efficiencies. In the paper, input variables included the use of rice bran, mustard oil cake, lime, labor, and seeds stocked, while output variables were total revenue from fish production. The findings were used to develop methods to assist farmers with improving their farm management.

Considering the findings of the paper, the estimated technical efficiency value for Jammu and Kashmir was highest of 0.9771 and 0.9741, respectively, with the lowest technical inefficiency of 3% The mean allocative and cost efficiencies were also estimated, revealing that the farmers in both regions were more allocative efficient than cost efficient. This higher allocative efficiency means that farmers were able to produce maximum possible output at minimum costs. We found that farming experience and years of education were the important factors, which influenced the efficiency in the farms, and the negative coefficient of household size indicates that as the size of the family grows, more produce is consumed at home, reducing the percentage of the quantity actually sold in the market.

There is a need for the government to promote the fish culture in the state as the current exotic culture of grass carp, common carp, and silver carp is scattered over the area leading to economic inefficiencies. Policymakers can help farmers improve their farm management by focusing on farmers with lower levels of education, fewer years of

experience, and less capital. Fish farmer specialized cooperatives should be developed to bring farmers together so that exchange of experience and positive discussion amongst them could help in increasing the efficiencies of the new farmers and promote the fish culture in the area.

**Author Contributions:** H.Z.: Conceptualization, methodology, software, writing—original draft preparation: U.N. and Y.M.: Visualization, investigation, supervision. All authors have read and agreed to the published version of the manuscript.

**Funding:** The authors acknowledge the financial support from the China Agriculture Research System of MOF and MARA.

**Data Availability Statement:** Not applicable.

**Conflicts of Interest:** The authors declare no conflict of interest. The funders had no role in the design of the study; in the collection, analyses, or interpretation of data; in the writing of the manuscript, or in the decision to publish the results.

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
