# Peer review of "Evaluation of Technical Efficiency in Exotic Carp Polyculture in Northern India: Conventional DEA vs. Bootstrapping Methods"

_fishes, doi:10.3390/fishes7040168_

Round 1

Reviewer 1 Report

The authors made an interesting evaluation of technical efficiency in the polyculture of exotic carp in northern India, using conventional methods DEA and bootstrapping. Despite the good presentation, there are a lot of errors that need to be corrected.

References must be used in accordance with the instructions for authors.

Example, line 115. [15] applied DEA..correct in Long et al [15] used DEA, similar to line 147: [25]; line 230: [30]; 275: [34]; line 424, line 427, line 452, line 457, line 460

Also reference in lines 261 and 271.

Lines 75 and 174 Dara Envelopment Analysis (DEA)

Spacing in text must be checked and consistent, as in lines 34 and 197, etc.

Reviewer 2 Report

The study is an application of technical efficiency analysis in the sector of exotic carp polyculture in India. A Data Envelopment Analysis approach is compared with a Bootstrapping method. Both methods are well run and results are clearly showed. However, I suggest some adjustments in order to improve the quality of the paper:

1. It is not clear why DEA and Bootstrapping methods are compared. What is the reason of comparing that models and not others (e.g., parametric methods)?

2. The authors select four variables that could affect inefficiency. What is the rationale of this choice? What are the theoretical reasons at the basis of this choice?

3. The TE estimated with the DEA is really high. What does imply in terms of homogeenity of the sample?

4. Generally, when different (fish or other) species are grown the farms can significantly differ in terms of techniques and technologies used. DEA allows possibility to estimate if techological differences exist among farms and, as a consequence, if farms lie on a unique or on separated frontiers. The authors should estimate that.

Author Response

This manuscript is a resubmission of an earlier submission. The following is a list of the peer review reports and author responses from that submission.